# Measurement and Calculation of the Impedance of an Eddy Current Probe Placed Above a Disc with Two Layers of Different Diameters

**DOI:** 10.3390/ma18102376

**Published:** 2025-05-20

**Authors:** Yike Xiang, Grzegorz Tytko, Yao Luo, Jolanta Makowska

**Affiliations:** 1State Key Laboratory of Power Grid Environmental Protection, School of Electrical Engineering and Automation, Wuhan University, Wuhan 430072, China; 2School of Electrical Engineering and Automation, Wuhan University, Wuhan 430072, China; 3Faculty of Automatic Control, Electronics and Computer Science, The Silesian University of Technology, Akademicka 16, 44-100 Gliwice, Poland; 4Institute of Materials Engineering, Faculty of Science and Technology, University of Silesia, 75 Pułku Piechoty 1A, 41-500 Chorzow, Poland

**Keywords:** eddy current testing, probe impedance, conductive disc, truncated region eigenfunction expansion (TREE), analytical model, coil resistance, coil reactance

## Abstract

This work presents a system developed to determine changes in the impedance of an eddy current probe placed above a conductive disc containing two layers of different diameters. In the first step, an analytical model was derived with an employment of the truncated region eigenfunction expansion (TREE) method. The final formula for the probe impedance change was presented in a closed form, which makes it possible to implement it in any programming language or computer algebra system. The mathematical model was implemented in MATLAB and used to design probes and to determine the optimal test parameters. In the next step, two eddy current probes with a single coil with different geometric dimensions were constructed. Impedance measurements were carried out using an LCR meter for three sets of double-layer discs. The tested discs were made of materials with different electrical conductivities. The upper and lower layers of the disc also differed in terms of the geometric dimensions, i.e., the diameter and thickness. The tests were performed for the operating frequency of the probe ranging from 1 kHz to 10 kHz. In all cases, a very good agreement was obtained between the measurement and the calculation results. Both the error in the changes in resistance and the error in the changes in reactance did not exceed 3.5%.

## 1. Introduction

The eddy current technique is one of the most commonly used methods of the non-destructive testing of electrically conductive materials. The inspection is carried out using an eddy current probe, which usually contains one [1,2,3,4,5], two [6,7,8,9], or more coils [10,11,12,13,14,15,16]. The principle of operation of the probe is based on the phenomenon of electromagnetic induction. The coil powered by an alternating current generates a magnetic field, which, when brought close to a conductive material, induces eddy currents in it. The occurrence of corrosion, material degradation, or structural changes, such as cracks, slots, and delaminations, disturbs the flow of eddy currents, resulting in a change in the probe impedance as a consequence. A precise measurement of impedance changes and their correct interpretation make it possible to determine the technical condition of the tested object. An analysis of the obtained changes in the impedance of the probes also allows us to determine the geometrical dimensions of the test material (thickness, width, and radius) [17,18,19,20,21], its electrical conductivity [22,23,24,25], and it magnetic permeability [26,27,28].

Comprehensive eddy current tests are carried out using systems consisting of a measuring device (a probe with an impedance meter) and software with an implemented mathematical model. Many tests have a mathematical model as an integral part of the measurement system, because it allows for a correct interpretation of the measured impedance values. By performing calculations for different values of the tested material parameters, it is possible to determine their impact on the inspection result and to point to the causes of the impedance changes obtained during the measurement. In this way, it is also possible to perform computer simulations and determine the optimal inspection parameters, such as the operating frequency of the probe or the distance of the probe from the tested surface (lift-off).

Numerical mathematical models usually utilise mesh methods, such as the finite element method (FEM) [29,30] or the boundary element method (BEM) [31,32]. The creation of numerical models is not time-consuming; therefore, they are often used for eddy current problems with complex geometry. However, because of the relatively long time of calculations, in the case of problems requiring many thousands of iterations, analytical models are employed. Analytical solutions are based on precomputation, where the only equations solved in each iteration are those within which the values of the input parameters have changed. Moreover, the final formulae in a closed form allow for their implementation in any programming language or computational package, and do not require a high computing power. Unfortunately, the creation of analytical models is very difficult, and, thus far, the developed and presented solutions only pertain to certain eddy current problems.

A milestone in the analytical modelling of eddy current problems was the formulation of integral expressions for the change in the impedance of a coil placed above a conducting half-space, as presented by Dodd and Deeds [33,34]. The development of the truncated region eigenfunction expansion (TREE) method by T. Theodoulidis resulted in obtaining the final expressions in the form of the sum of the series [35,36]. New find-root algorithms [37,38,39] paved the way for modelling multilayer conductors containing defects [40,41,42], as well as more complex elements, such as rods or tubes [43]. Next, analytical solutions for single-layer [44] and multilayer [45] discs were obtained. Regrettably, in all the models created so far, each layer of a small conductor was supposed to have the same diameter. This limitation was the main obstacle to the further development of analytical modelling, obstructing the modelling of many essential products, such as screws or rivets. This paper introduces a solution to this problem that has never been presented before. The developed system consists of a constructed eddy current probe, an LCR meter, and an analytical model derived with the employment of the TREE method. The implementation of the obtained final formulas in MATLAB R2024b made it possible for us to carry out a rapid calculation of the impedance of the probe placed above the disc with two layers of different diameters. Impedance measurements were performed using two eddy current probes, each containing a single coil with a different number of turns and distinct geometric dimensions. Three sets of double-layer discs made of conductive materials were subjected to testing, with each layer having a different diameter, thickness, and electrical conductivity. In all cases, a good agreement was obtained for both the coil resistance and reactance, proving the effectiveness of the developed measurement system.

## 2. Materials and Methods

### 2.1. Mathematical Model

A two-dimensional analytical model was derived using the TREE method. According to the assumptions of this method, the infinite solution domain is limited in the radial direction *r* to an arbitrarily adopted value. Usually, this value is the multiple of the outer radius of the coil. Owing to this, in calculations carried out with the employment of the TREE model, it is possible to utilise coils of different geometrical dimensions and yet avoid significant numerical errors. Then, the solution domain in the axial direction (along axis *z*) is divided into regions with different electrical properties (electrical conductivity), magnetic properties (magnetic permeability), or geometrical dimensions. Regions consisting solely of air are usually located above the coil, between the coil and the tested material, and below the tested material. In the case of the cored coil—the utilised core contains 1 region (I-core [4,46,47]), 2 regions (T-core [48] and C-core [49]), or 3 regions (E-core [1,50,51,52,53]). Most often, each layer of the tested material also constitutes a separate region. Once all regions have been defined, for each of them, an expression for the magnetic vector potential is determined. These expressions comprise eigenvalues, the calculation of which requires defining subregions. Regions that do not have a uniform structure are divided into subregions with different properties (for example, core, air, and conductive material). The comparison of the expressions for the magnetic vector potential of the neighbouring subregions allows for the formulation of a system of equations. The solution of this system is a set of eigenvalues. In the subsequent step, the interface equations are defined using the conditions of magnetic field continuity at the interfaces between the regions. Depending on the geometry of the problem under consideration, the number of interface equations varies from a few to a dozen or so. The solution of this system makes it possible to determine all unknown coefficients in the expressions for the magnetic vector potential of each region. What is to be carried out in the last step is a double integration of the expression for the magnetic potential of the region with the coil, in order to obtain the final formula for the change in the coil impedance.

The analysed problem was treated as axisymmetric and considered in a cylindrical coordinate system (Figure 1). The solution domain was divided into 4 regions (1–4) and limited in the radial direction to the value of parameter *a*. A coil with height *z*_2_−*z*_1_ and width *r*_2_–*r*_1_ was placed parallel at a distance *z*_1_ from the surface of the conductive disc. The electrical conductivity of the upper layer of the disc with thickness *b*_1_ and radius *a*_1_ was *σ*_1_. Analogously, the electrical conductivity of the lower layer of the disc with thickness *b*_2_ and radius *a*_2_ was *σ*_2_. In the first step, the expressions for the magnetic vector potential *A*(*r*, *z*) of each region were written in matrix form:(1)$$A_{1}\left(r,z\right)=J_{1}\left(\alpha r\right)\left(e^{zA}C^{\left(e\right)}+e^{-zA}D_{1}\right),$$(2)$$A_{2}\left(r,z\right)=F_{1}\left(p_{1}r\right)\left(e^{zP_{1}}C_{2}+e^{-zP_{1}}D_{2}\right),$$(3)$$A_{3}\left(r,z\right)=F_{2}\left(p_{2}r\right)\left(e^{zP_{2}}C_{3}+e^{-zP_{2}}D_{3}\right),$$(4)$$A_{4}\left(r,z\right)=J_{1}\left(\alpha r\right)e^{zA}C_{4},$$
where(5)$$A=\left[\begin{array}{ccc} \alpha_{1} & 0 & \ldots \\ 0 & \alpha_{2} & \ldots \\ \vdots & \vdots & \ddots \end{array}\right],\;P_{1}=\left[\begin{array}{ccc} p_{1,1} & 0 & \ldots \\ 0 & p_{1,2} & \ldots \\ \vdots & \vdots & \ddots \end{array}\right],\;P_{2}=\left[\begin{array}{ccc} p_{2,1} & 0 & \ldots \\ 0 & p_{2,2} & \ldots \\ \vdots & \vdots & \ddots \end{array}\right],$$(6)$$C^{\left(e\right)}=\frac{1}{2}\left(e^{-Az_{1}}-e^{-Az_{2}}\right)\frac{2\mu_{0}}{\alpha^{2}a^{2}J_{0}^{2}(\alpha a)}\kappa (\alpha ),$$(7)$$\kappa (\alpha )=\int_{r_{1}}^{r_{2}}rJ_{1}\left(\alpha r\right)dr,$$(8)$$F_{1}\left(p_{1}r\right)=\left\{\begin{array}{c} J_{1}\left(q_{1}r\right)R_{1}\left(p_{1}a_{1}\right),\;\;\;\;\;\;\;\;0\leq r\leq a_{1}\\ J_{1}\left(q_{1}a_{1}\right)R_{1}\left(p_{1}r\right),\;\;\;\;\;\;\;\;\;a_{1}\leq r\leq a \end{array}\right.,$$(9)$$F_{2}\left(p_{2}r\right)=\left\{\begin{array}{c} J_{1}\left(q_{2}r\right)R_{1}\left(p_{2}a_{2}\right),\;\;\;\;\;\;\;\;\;\;\;0\leq r\leq a_{2}\\ J_{1}\left(q_{2}a_{2}\right)R_{1}\left(p_{2}r\right),\;\;\;\;\;\;\;\;\;\;\;a_{2}\leq r\leq a \end{array}\right.,$$(10)$$R_{n}\left(pr\right)=J_{n}\left(pr\right)Y_{1}\left(pa\right)-J_{1}\left(pa\right)Y_{n}\left(pr\right),$$(11)$$q_{n}=\sqrt{p_{n}^{2}-j\omega \mu_{0}\sigma_{n}},$$

*J*(*x*) and *Y*(*x*) are Bessel functions of the first and second kind, *ω* is the angular frequency, *μ*_0_ is the permeability of free space, and **C**_2_–**C**_4_ and **D**_1_–**D**_3_ are unknown coefficients. Regions 1 and 4 are uniform and consist entirely of air. Their eigenvalues *α* are real numbers that can be calculated by solving the equation *J*_1_(*α a*) = 0. Regions 2 and 3 consist of two subregions. Region 2 contains the upper layer of the conductive disc (0 ≤ *r* ≤ *a*_1_) and air (*a*_1_ ≤ *r* ≤ *a*). Its complex eigenvalues *p*_1_ are roots of the function:(12)$$\chi_{1}\left(p\right)={p_{1}J}_{1}\left(q_{1}a_{1}\right)R_{0}\left(p_{1}a_{1}\right)-{q_{1}R}_{1}\left(p_{1}a_{1}\right)J_{0}\left(q_{1}a_{1}\right).$$

Region 3 consists of the lower layer of the disc (0 ≤ *r* ≤ *a*_2_) and air (*a*_2_ ≤ *r* ≤ *a*). The eigenvalues *p*_2_ of region 4 are complex roots of the function:(13)$$\chi_{2}\left(p\right)={p_{2}J}_{1}\left(q_{2}a_{2}\right)R_{0}\left(p_{2}a_{2}\right)-{q_{2}R}_{1}\left(p_{2}a_{2}\right)J_{0}\left(q_{2}a_{2}\right).$$

Taking into account the following conditions of magnetic field continuity for the neighbouring regions,(14)$$A_{i}(r,z)=A_{i+1}(r,z),$$(15)$$\frac{\partial A_{i}}{\partial z}=\frac{\partial A_{i+1}}{\partial z},$$
a system of six interface equations, Equations (16)–(21), was obtained.(16)$$S\left[C^{\left(e\right)}+D_{1}\right]=T_{1}\left[C_{2}+D_{2}\right],$$(17)$$T_{1}\alpha \left[C^{\left(e\right)}-D_{1}\right]=N_{1}p_{1}\left[C_{2}-D_{2}\right],$$(18)$$N_{1}\left[{e^{b_{1}p_{1}}C}_{2}+{e^{{-b}_{1}p_{1}}D}_{2}\right]=U\left[{e^{b_{1}p_{2}}C}_{3}+{e^{{-b}_{1}p_{2}}D}_{3}\right],$$(19)$$N_{1}p_{1}\left[{e^{b_{1}p_{1}}C}_{2}-{e^{{-b}_{1}p_{1}}D}_{2}\right]=Up_{2}\left[{e^{b_{1}p_{2}}C}_{3}-{e^{{-b}_{1}p_{2}}D}_{3}\right],$$(20)$$T_{2}\left[{e^{b_{2}p_{2}}C}_{3}+{e^{{-b}_{2}p_{2}}D}_{3}\right]=S{e^{b_{2}A}C}_{4},$$(21)$$T_{2}p_{2}\left[{e^{b_{2}p_{2}}C}_{3}-{e^{{-b}_{2}p_{2}}D}_{3}\right]=S\alpha {e^{b_{2}A}C}_{4},$$
where(22)$$S=\int_{0}^{a}rJ_{1}(r)J_{1}(r)dr,$$(23)$$T_{n}=\int_{0}^{a}rJ_{1}(r)F_{n}(r)dr,$$(24)$$N_{1}=\int_{0}^{a}rF_{1}(r)F_{1}(r)dr,$$(25)$$U=\int_{0}^{a}rF_{1}(r)F_{2}(r)dr.$$

Writing the system of Equations (16)–(21) in the form *AX* = *B*, the following coefficient matrices were obtained:(26)$$\left[\begin{array}{ccc} A_{11} & A_{12} & A_{13}\\ A_{21} & A_{22} & A_{23}\\ A_{31} & A_{32} & A_{33} \end{array}\right]\left[\begin{array}{c} C_{2}\\ D_{2}\\ C_{3} \end{array}\right]=\left[\begin{array}{c} 2T_{1}\alpha C^{\left(e\right)}\\ 0\\ 0 \end{array}\right].$$

Solving (26), coefficients **C**_2_, **D**_2_, and **D**_3_ were obtained, and, subsequently, the coefficient **D**_1_ was determined.(27)$$D_{1}={S^{-1}T}_{1}\left[C_{2}+D_{2}\right]-C^{\left(e\right)}.$$

The final formula for the change in the impedance of a coil placed above a conductive disc with two layers of different diameters was written in the following form:(28)$$\Delta Z=\frac{j\omega 2\pi N^{2}}{(r_{2}-r_{1}{)}^{2}(z_{2}-z_{1}{)}^{2}}\sum_{j=1}^{\infty }\kappa (\alpha )\frac{e^{-\alpha_{j}z_{2}}-e^{-\alpha_{j}z_{1}}}{\alpha_{j}}D_{1,j}.$$

### 2.2. Measurement System

For the purposes of the conducted tests, two eddy current probes with a single coil (Figure 2), whose parameters are presented in Table 1, were designed and constructed. All dimensions of the coils were measured with a calliper with an accuracy of ±0.1 mm. The values of the internal and external diameters of both coils were similar. The remaining parameters, i.e., the wire diameter, the number of turns, and the height of the coil, were made significantly different in order to investigate their effect on the probe sensitivity. Coil C1 was wound with thick wire, obtaining a significant height of 15.8 mm with a relatively small number of turns. Coil C2 was wound with very thin wire, which resulted in about three times more turns and three times smaller height than in coil C1.

The measurement stand is shown in Figure 3. The probes were connected to the Keysight E4980A precision LCR meter using clip leads (Santa Rosa, CA, USA). The measurement results were displayed in real time on a personal computer with Keysight BenchVue 2024 software installed. The measurements of the components of the probe impedance were performed for three sets of double-layer discs with an accuracy of ±0.05%. The geometric dimensions and electrical conductivity values of the discs shown in Figure 2 were recorded in Table 2. The geometric dimensions of the discs were measured with a micrometer screw with an accuracy of ±0.01 mm, whereas the electrical conductivity of the materials the discs were made of was determined using a Foerster Sigmatest 2.069 tester with an accuracy of ±0.5% of the measured value (Foerster, Reutlingen, Germany). The sets of discs were selected in such a manner that they were characterised by a large diversity. In the first set, the upper layer is much thicker than the lower layer and has a higher electrical conductivity value. The second set shows the reverse relationship; i.e., both the thickness and electrical conductivity of the upper layer are smaller than those of the lower layer. The last set has a configuration in which the upper layer made of graphite is much thicker and has a much lower electrical conductivity than the lower layer made of bronze.

## 3. Results and Discussion

The final Formula (28) for the change in the coil impedance was implemented in MATLAB. The complex eigenvalues *p*_1_ and *p*_2_ were calculated with the employment of the root-finding algorithm developed in [43]. The integral (7) was calculated using the sum of the series employing 20 terms:(29)$$\kappa (\alpha )=\frac{1}{\alpha }[r_{1}J_{0}(\alpha r_{1})-r_{2}J_{0}(\alpha r_{2})]+\frac{2}{\alpha^{2}}\sum_{n=0}^{\infty }[J_{2n+1}(\alpha r_{2})-J_{2n+1}(\alpha r_{1})].$$

The calculations and measurements were performed for 50 frequency values within the range of 1 kHz to 10 kHz, which was selected to ensure that the eddy current penetration depth would be sufficient to fully penetrate both disk layers. At low frequencies (below 1 kHz), the impedance component values become very small, which can result in relatively larger errors in the calculated results. However, no significant increase in error was observed for frequencies above 10 kHz. Based on these observations, the model is considered reliable within the selected frequency range. First, impedance *Z*_0_ = *R*_0_ + *j X*_0_ was measured for the coil located in a space without conductive material. Then, impedance Z = *R* + *j X* was measured for the coil placed on the surface of the conductive disc. The change in impedance Δ*Z* was determined as the difference of the obtained values, i.e., Δ*Z* = *Z* − *Z*_0_. Each measurement of the impedance components was performed eight times, and, subsequently, the arithmetic mean of the measured resistance and reactance values was calculated. The coils were equipped with 0.4 mm (coil C1)- and 0.2 mm (coil C2)-thick pads to obtain the different distances between the lower edge of the probe and the surface of the tested material (lift-off). In order to compare the results achieved with the measurement and mathematical model, the relative error Δ*R* of the resistance and the relative error Δ*X* of the reactance were defined and expressed in percentages.(30)$$\delta R=\left|\frac{\Delta R_{measured}-\Delta R_{calculated}}{\Delta R_{measured}}\right|\cdot 100\%$$(31)$$\delta X=\left|\frac{\Delta X_{measured}-\Delta X_{calculated}}{\Delta X_{measured}}\right|\cdot 100\%$$

In the first series, a set of one double-layer disc consisting of brass (upper layer) and bronze CC481K (bottom layer) was tested. The obtained changes in resistance Δ*R* = *R* − *R*_0_ and reactance Δ*X* = *X* − *X*_0_ for coils C1 and C2 are presented in Table 3. The values of the impedance components obtained using the analytical TREE model and measurement were compared using the coefficients Δ*R* and Δ*X*. In the next series, sets of discs 2 and 3 were tested. The results of the calculations and measurements are presented in Table 4 and Table 5.

In the next step, changes in the resistance Δ*R* of coil C1, obtained for sets 1–3, were normalised with respect to reactance *X*_0_, and presented in Figure 4. The calculation results are connected with a continuous line, and each of the 50 measurement results is presented as a circle. The normalised changes in the reactance Δ*X* of coil C1 are shown in Figure 5. The same procedure was followed for coil C2—the obtained results after normalisation with respect to reactance *X*_0_ are presented in Figure 6 and Figure 7. The changes in the impedance components were normalised with reference to reactance *X*_0_ in order to make the obtained results independent of the geometrical dimensions of the coils. Moreover, the use of this approach, which is commonly employed in eddy current testing, makes it possible to compare the results published in different articles.

In the entire frequency range under consideration, the resistance error Δ*R* and the reactance error Δ*X* were less than 3.5% for all 50 measurement points. This value is much lower than 5%, which is acceptable by the authors, and which is usually accepted as the allowable difference between the results of calculations and measurements in this type of eddy current inspections. Such a good agreement of the obtained changes in the resistance and reactance evidences the correctness of the derived analytical model and its correct numerical implementation. At the same time, the calculation results confirm the correct configuration of the applied measurement system, including the design of the eddy current probes, calibration of the LCR meter, and measurement of all geometric dimensions of the probes and discs.

Based on the results in Figure 4, Figure 5, Figure 6 and Figure 7, it was found that the values of the changes in the impedance components obtained for sets 2 and 3 were much smaller than for set 1. In the case of sets 2 and 3, the difference in the changes in the normalised resistance is the biggest for low frequencies. It is then that the penetration depth of the eddy current is the greatest, and these sets differ primarily in the electrical conductivity of the lower layer. Along with the increasing frequency, when the penetration depth decreases, the difference between the normalised Δ*R* values gradually decreases. This correlation shows how significant a correct selection of the operating frequency of the probe is. In the case of *f* = 6 kHz for coil C1 and *f* = 8 kHz for coil C2, the changes in the probe resistance are the same, which makes it impossible to distinguish between sets 2 and 3 based on the analysis of the obtained values of this parameter.

In the case of each set, the changes in the impedance components obtained for coil C2 were significantly greater than those for coil C1. The parameters that determined this fact were a shorter lift-off distance (0.2 mm for C2 and 0.4 mm for C1) and a more than three times greater number of turns (1650 for C2 and 480 for C1). The height of the coil (15.8 mm for C1 and 5.4 mm for C2) only slightly affected the changes in the impedance. The significantly higher coil C1 also had a much larger winding area (164.32 mm2) than coil C2 (57.24 mm2). Despite this, the value of the changes in the resistance and reactance obtained for coil C1 were many times smaller than those in the case of coil C2.

The sensitivity of the developed mathematical model to changes in the input parameters was studied by changing the following values sequentially:-The inner radius of the coil *r*_1_;-The outer radius of the coil *r*_2_;-The distance of the coil from the conducting disc (lift-off) *z*_1_;-The height of coil *z*_2_−*z*_1_;-The electrical conductivity *σ*_1_;-The thickness of the upper disc layer *b*_1_.

In all tests, which were performed at a frequency of *f* = 10 kHz, the value of only one parameter was changed at a time. The obtained calculation results are presented in Table 6 and Table 7, and the following observations are made.

Increasing the inner radius *r*_1_ or outer radius *r*_2_ of the coil led to an increase in both the real part and the absolute value of the imaginary part of the impedance change, while increasing the coil height (*z*_2_−*z*_1_) led to an opposite change. These are attributed to the change in the overall shape and size of the coil. Increasing the distance between the coil and the discs (by increasing *z*_1_) resulted in a decrease in both the real part and the absolute value of the imaginary part of the impedance change, due to the weaker electromagnetic interaction between the coil and the discs.

Increasing the electrical conductivity caused the real part of the impedance change to decrease, while the absolute value of the imaginary part increased.

Increasing the thickness of the disc initially caused the real part of the impedance change to decrease, followed by a slow increase, whereas the absolute value of the imaginary part exhibited the opposite trend.

## 4. Conclusions

The measurement system proposed in this paper constitutes an analytical solution to the problem of an eddy current probe placed above a conductive disc with two layers of different diameters. The developed solution enables effective measurements of the probe impedance during eddy current inspections of elements of this type. In this way, it is possible to examine the technical condition of conductive discs and monitor their parameters, such as geometric dimensions or electrical conductivity. The created mathematical model may be used to design probes, determine the optimal inspection parameters, carry out computer simulations of the tests, and, above all, interpret the measurement results.

The measurement results were compared with the results of the calculations with the employment of two eddy current probes designed by the authors. The probes differed in the geometric dimensions, number of turns, and value of the lift-off distance. The tests were carried out for three sets of discs made of materials with different electrical conductivities. In addition, the disc layers had different diameters and different thicknesses. In all tests, the errors in resistance and reactance changes were less than 3.5%. Such a good agreement between the measurement and calculation results confirms the correctness of the proposed solution and facilitates starting work on more complex geometries. Further research will aim to extend the analytical solutions for probes consisting of two coils operating in a transmitter–receiver configuration, and for probes containing ferrite cores. It is also planned to take into consideration various inhomogeneities of the conductive material, such as delaminations, material inclusions, and cracks. The main objective of the further development of the mathematical model is to derive a 3D solution which will enable us to solve problems that cannot be regarded as axisymmetric.

## Figures and Tables

**Figure 1 materials-18-02376-f001:**
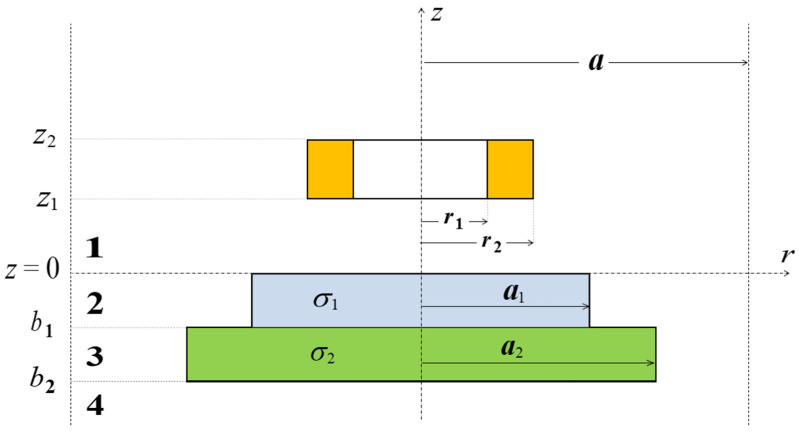
Cross-section of a coil placed above a conductive disc with two layers of different diameters.

**Figure 2 materials-18-02376-f002:**
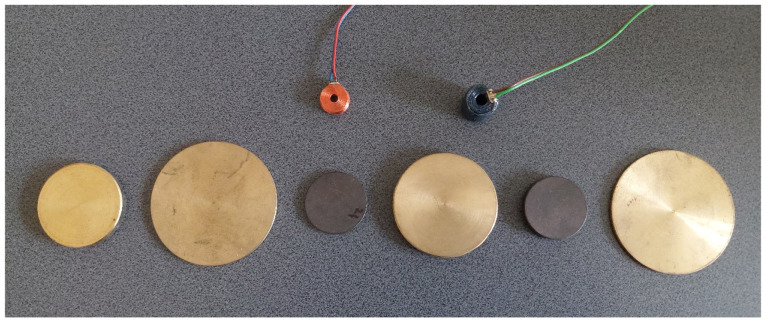
Coils and discs used in the experiments.

**Figure 3 materials-18-02376-f003:**
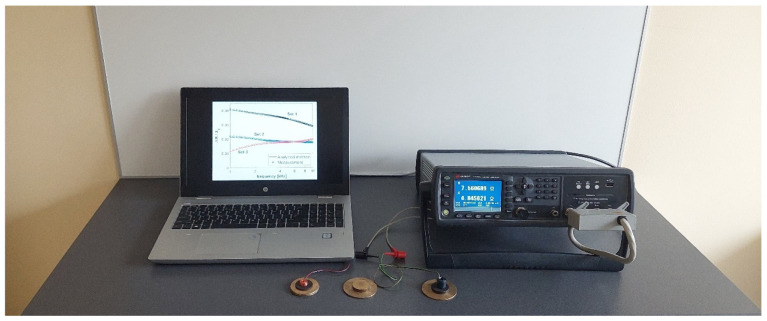
Measuring station with the precision LCR meter Keysight E4980A, personal computer, probes, and discs.

**Figure 4 materials-18-02376-f004:**
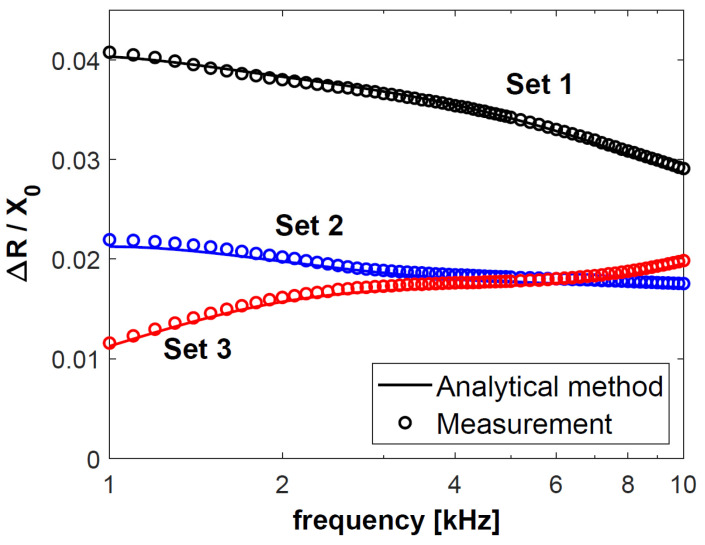
Normalised changes in the resistance Δ*R* of coil C1 for sets 1–3 of double-layer discs.

**Figure 5 materials-18-02376-f005:**
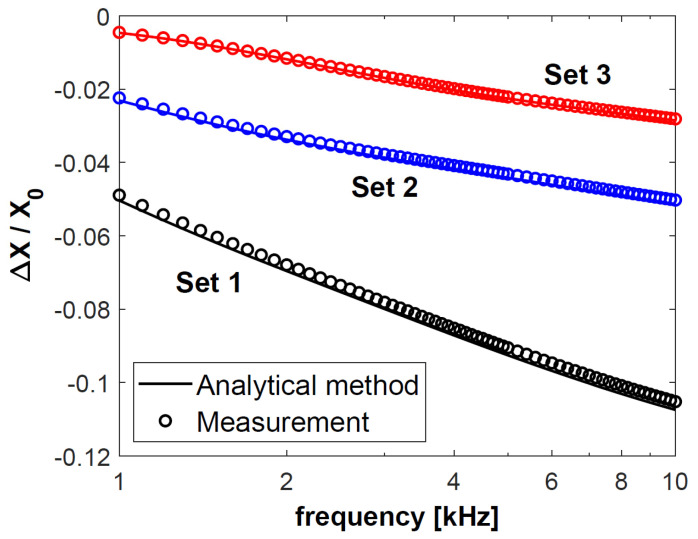
Normalised changes in the resistance Δ*X* of coil C1 for sets 1–3 of double-layer discs.

**Figure 6 materials-18-02376-f006:**
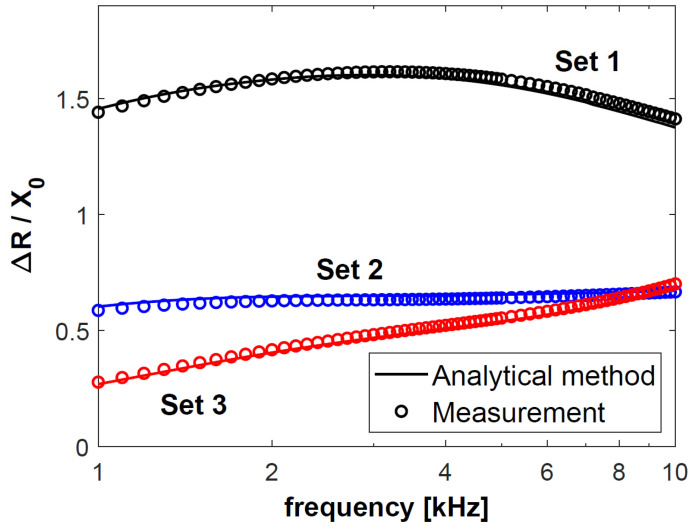
Normalised changes in the resistance Δ*R* of coil C2 for sets 1–3 of double-layer discs.

**Figure 7 materials-18-02376-f007:**
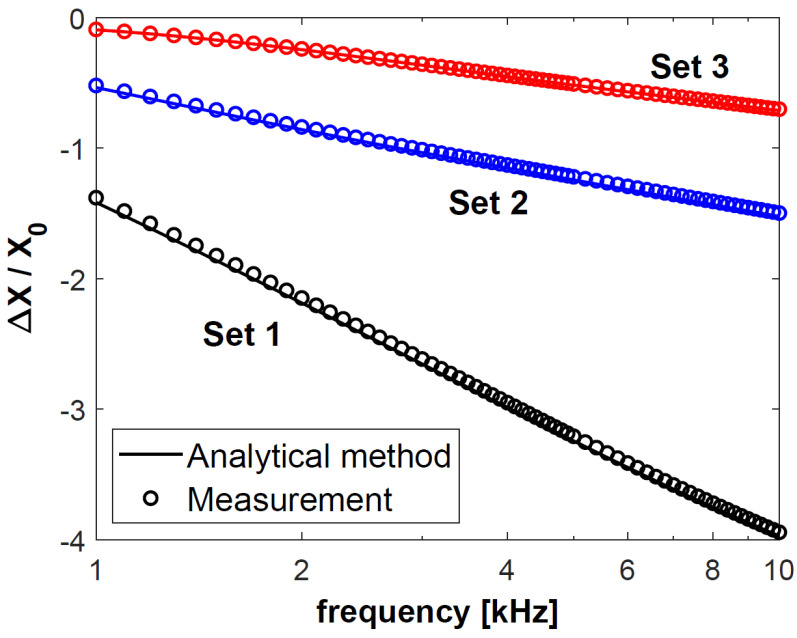
Normalised changes in the resistance Δ*X* of coil C2 for sets 1–3 of double-layer discs.

**Table 1 materials-18-02376-t001:** Parameters of the coils used in the experiments.

Parameter	Symbol	Coil C1	Coil C2
Number of turns	* N *	480	1650
Inner radius	* r * _1_	2.6 mm	2.0 mm
Outer radius	* r * _2_	7.8 mm	7.3 mm
Height	*z*_2_−*z*_1_	15.8 mm	5.4 mm

**Table 2 materials-18-02376-t002:** Parameters of the two-layer discs.

Set	Layer	Conductive Material	Radius [mm]	Thickness [mm]	Electrical Conductivity [MS/m]
1	Upper	Brass	19.97	4.83	14.22
1	Lower	Bronze CC481K	29.81	2.07	6.09
2	Upper	Graphite	15.01	2.13	0.45
2	Lower	Bronze CW452K	25.21	5.04	9.48
3	Upper	Graphite	15.08	5.02	0.45
3	Lower	Bronze CC481K	29.79	2.04	6.09

**Table 3 materials-18-02376-t003:** Changes in the impedance components of coil C1 and C2 obtained for set 1.

		Measurement		Analytical Method			
Coil	*f* [kHz]	Δ*R* [Ω]	Δ*X* [Ω]	Δ*R* [Ω]	Δ*X* [Ω]	Δ*R* [%]	Δ*X* [%]
C1	1	0.212	−0.255	0.210	−0.262	0.94	−2.75
C1	5	0.890	−2.355	0.888	−2.408	0.22	−2.25
C1	10	1.513	−5.473	1.490	−5.590	1.52	−2.14
C2	1	7.49	−7.18	7.57	−7.36	−1.07	−2.51
C2	5	41.18	−83.41	40.50	−84.33	1.65	−1.10
C2	10	73.43	−205.09	71.50	−206.59	2.63	−0.73

**Table 4 materials-18-02376-t004:** Changes in the impedance components of coil C1 and C2 obtained for set 2.

		Measurement		Analytical Method			
Coil	*f* [kHz]	Δ*R* [Ω]	Δ*X* [Ω]	Δ*R* [Ω]	Δ*X* [Ω]	Δ*R* [%]	Δ*X* [%]
C1	1	0.114	−0.117	0.111	−0.120	2.63	−2.56
C1	5	0.473	−1.123	0.461	−1.149	2.54	−2.32
C1	10	0.912	−2.615	0.892	−2.672	2.19	−2.18
C2	1	3.05	−2.70	3.13	−2.79	−2.62	−3.33
C2	5	16.64	−31.71	17.01	−32.34	−2.22	−1.99
C2	10	34.64	−77.87	34.88	−79.20	−0.69	−1.71

**Table 5 materials-18-02376-t005:** Changes in the impedance components of coil C1 and C2 obtained for set 3.

		Measurement		Analytical Method			
Coil	*f* [kHz]	Δ*R* [Ω]	Δ*X* [Ω]	Δ*R* [Ω]	Δ*X* [Ω]	Δ*R* [%]	Δ*X* [%]
C1	1	0.060	−0.024	0.059	−0.024	1.67	0.00
C1	5	0.462	−0.576	0.451	−0.587	2.38	−1.91
C1	10	1.032	−1.463	1.017	−1.485	1.45	−1.50
C2	1	1.44	−0.47	1.40	−0.48	2.78	−2.13
C2	5	14.39	−13.20	14.02	−13.44	2.57	−1.82
C2	10	36.42	−36.46	35.70	−36.95	1.98	−1.34

**Table 6 materials-18-02376-t006:** Changes in the impedance components of coil C1 obtained for different values of parameters *r*_1_, *r*_2_, *z*_1_.

*r*_1_ [mm]	Δ*R* [Ω]	Δ*X* [Ω]	*r*_2_ [mm]	Δ*R* [Ω]	Δ*R* [Ω]	*z*_1_ [mm]	Δ*R* [Ω]	Δ*X* [Ω]
2.0	1.27	−4.74	6.0	0.95	−3.16	0.0	1.82	−6.55
2.2	1.34	−5.01	6.2	1.01	−3.38	0.4	1.49	−5.59
2.4	1.41	−5.29	6.4	1.07	−3.62	0.8	1.23	−4.80
2.6	1.49	−5.59	6.6	1.13	−3.86	1.2	1.02	−4.16
2.8	1.57	−5.90	6.8	1.19	−4.12	1.6	0.85	−3.61
3.0	1.65	−6.23	7.0	1.25	−4.39	2.0	0.72	−3.16
3.2	1.73	−6.57	7.2	1.31	−4.67	2.4	0.61	−2.78
3.4	1.81	−6.93	7.4	1.37	−4.97	2.8	0.52	−2.45
3.6	1.90	−7.30	7.6	1.43	−5.27	3.2	0.45	−2.17
3.8	1.99	−7.69	7.8	1.49	−5.59	3.6	0.38	−1.93
4.0	2.07	−8.10	8.0	1.55	−5.92	4.0	0.33	−1.72
5.0	2.54	−10.36	9.0	1.87	−7.73	6.0	0.17	−1.02
6.0	3.03	−13.01	10.0	2.21	−9.80	8.0	0.10	−0.63

**Table 7 materials-18-02376-t007:** Changes in the impedance components of coil C1 obtained for different values of parameters *z*_2_−*z*_1_, *σ*_1_, and *b*_1_.

*z*_2_−*z*_1_ [mm]	Δ*R* [Ω]	Δ*X* [Ω]	*σ*_1_ [MS/m]	Δ*R* [Ω]	Δ*R* [Ω]	*b*_1_ [mm]	Δ*R* [Ω]	Δ*X* [Ω]
10.0	3.29	−11.31	1	1.569	−2.108	0.4	1.97	−5.40
10.4	3.10	−10.73	5	1.890	−4.363	0.6	1.83	−5.57
10.8	2.92	−10.19	10	1.643	−5.222	0.8	1.72	−5.65
11.2	2.76	−9.69	15	1.466	−5.641	1.0	1.63	−5.69
11.6	2.61	−9.23	20	0.012	−7.698	2.0	1.46	−5.63
12.0	2.47	−8.79	25	0.011	−7.698	3.0	1.48	−5.59
12.4	2.34	−8.38	30	0.011	−7.697	4.0	1.49	−5.59
12.8	2.22	−8.01	35	0.010	−7.697	5.0	1.49	−5.59
13.2	2.11	−7.65	40	0.010	−7.697	6.0	1.49	−5.59
13.6	2.01	−7.32	45	0.010	−7.697	7.0	1.49	−5.59
14.0	1.91	−7.01	50	0.009	−7.697	8.0	1.49	−5.58
16.0	1.52	−5.70	55	0.009	−7.697	9.0	1.49	−5.58
18.0	1.24	−4.72	60	0.009	−7.697	10.0	1.49	−5.58

## Data Availability

The original contributions presented in this study are included in the article. Further inquiries can be directed to the corresponding author.

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
