# Peer review of "Measurement and Calculation of the Impedance of an Eddy Current Probe Placed Above a Disc with Two Layers of Different Diameters"

_materials, 2025, doi:10.3390/ma18102376_

Round 1

Reviewer 1 Report

Comments and Suggestions for Authors

This manuscript presents an analytical formulation based on the Truncated Region Eigenfunction Expansion (TREE) method for computing the impedance of an eddy current probe situated above a two-layer conductive disc with differing diameters. The authors derive a closed-form solution and validate the model against measurements taken from carefully prepared samples and hand-wound probes. The reported agreement, within 3.5% across a 1–10 kHz range, is encouraging.

The extension of the TREE framework to accommodate a geometry where the two layers have unequal radii is noteworthy. This class of problems has practical relevance in applications such as fastener inspection or evaluation of layered components with variable footprint — areas where traditional analytical models tend to fall short.

The mathematical framework appears sound and is well aligned with prior literature in this space. The derivation is methodical, and the use of closed-form expressions offers advantages in computational efficiency. Extending the TREE approach to this geometry is a meaningful contribution to the analytical modeling of eddy current problems.

The experimental validation uses two hand-wound coils and a limited set of discs. While suitable for initial validation, the scope could be broadened. Exploring a wider range of coil designs—ideally with tighter fabrication tolerances—would help demonstrate robustness. The influence of practical probe parameters (e.g., coil alignment, mutual coupling, lift-off repeatability) should also be considered.

The model assumes axisymmetry and ideal material homogeneity. These assumptions, while useful in deriving analytical expressions, limit the model’s applicability in field settings where asymmetry and material variability are common. It would be helpful to include a discussion on how the method might be extended or adapted to account for these effects.

The study focuses exclusively on nonmagnetic materials. In practice, many components of interest in NDE—such as ferrous alloys—have significant magnetic permeability. Including permeability in the model, or at least addressing how it might impact the formulation, would improve the utility of the approach.

The reported agreement between model and measurement is promising, but the absence of uncertainty quantification is a concern. Sensitivity of the model to input parameters such as conductivity, layer thickness, and coil geometry should be explored. This would strengthen confidence in the model’s predictive capability under realistic variations.

The model is validated over a 1–10 kHz range. Some discussion on how the TREE solution behaves near the limits of this band—particularly with respect to skin depth and field penetration—would add value. It would also be useful to understand if similar agreement holds at higher or lower frequencies.

Reviewer 2 Report

Comments and Suggestions for Authors

This manuscript presents an analytical and experimental study on impedance measurements of eddy current probes placed above double-layer conductive discs with differing diameters. The authors applied an analytical model based on the TREE (truncated region eigenfunction expansion) method and validate the results experimentally, using probes with varied geometric and electromagnetic properties. The comparison between model predictions and measurements demonstrates good agreement (the authors mentioned the the error rates are less than 3.5%), which supports the validity of the proposed methodology. The paper offers a valuable contribution to analytical modeling in eddy current testing, particularly in configurations with non-uniform disc geometries. I would like to say that the topic is of interest to researchers in materials characterization as well as electromagnetic modeling.

The concept of combining analytical modeling with experimental verification is strong and well-aligned with the scope of this journal. The introduction is well-structured and informative, clearly stating the existing gap in modeling layered discs with varying diameters. The methodology is generally presented, and the authors provide sufficient experimental details. However, several important issues should be addressed before the manuscript can be accepted:

1. The TREE method is central to the analytical part of the manuscript but is described rather briefly. A more detailed explanation of its application in Section 2 would benefit the reader.

2. Some notations in the equations and figures are inconsistent (e.g., incorrect mathematical symbols, missing italics).

3. There are minor linguistic inaccuracies that should be corrected.

4. Several numerical errors were identified in the reported results (Tables 3-5), which directly affect the integrity of Figures 4-7 and the validity of conclusions.

Here are my specific comments:

Line 47: Instead of "magnetically permeability", should be "magnetic permeability". Please revise.

Lines 84–88: Sentence structure is awkward and contains redundancy. Suggested revision:

"Impedance measurements were performed using two eddy current probes, each containing a single coil with a different number of turns and distinct geometric dimensions. Three sets of double-layer discs made of conductive materials were subjected to testing, with each layer having a different diameter, thickness, and electrical conductivity."

Line 92 (Figure 1): The labels are disproportionately large. Please reduce the font size for clarity.

Line 97: The authors are encouraged to describe the application of the TREE method in greater detail.

Lines 101 and 102: Instead of "was placed parallelly", should be "was placed parallel"

Lines 114 and 115: The variable "a" should be italicized in the inequalities to match the notation in Figure 1.

Line 152 (Equation 27): Please verify the mathematical formatting. The equation includes unusual symbols (e.g., a circled "a") which may have resulted from encoding issues. Confirm if the variable DZ contains a typo and should instead be ΔZ. The authors just give the Eq. (27) for change in the impedance. But how do they calculate the resistance and reactance parts?

Line 198 (Equation 28): Again, please check for symbol encoding errors (e.g., sum sign replaced with an unusual character).

Line 222 (Table 3): ΔR and ΔX values do not match the formulas provided in Eq. (29) and Eq. (30).
For example:
ΔR: 0.212 (meas) vs. 0.210 (analytical), error should be 0.94%, not 1.07%
ΔX: -0.255 vs. -0.262, error should be 2.74%, not 2.95%
Please, check all the other rows.

Lines 224 and 226 (Tables 4 and 5): Similar discrepancies are noted throughout these tables, except for the last two rows. The authors are advised to recalculate and verify all error values.

Lines 236, 239, 241, 245 (Figures 4-7): Since the tables contain incorrect ΔR and ΔX values, figures based on those values are also affected. Please verify the plotted values. Also, clarify why ΔR was normalized with X0 instead of R0.

Comments on the Quality of English Language

The manuscript is generally understandable, but there are several grammatical and stylistic issues mentioned in the comments that need correction. 

Round 2

Reviewer 1 Report

Comments and Suggestions for Authors

Taking into account the author's revisions, answers, and explanations, I think the manuscript is suitable for publication. 

Author Response

Thank you very much for your review.

Reviewer 2 Report

Comments and Suggestions for Authors

The authors have successfully addressed all the comments from the initial review, and the revised version demonstrates a significant improvement in clarity, structure, and analytical precision.

However, I would like to suggest a few additional minor corrections before final acceptance:

1. Lines 61–71: Please revise the sentence
“A milestone in analytical modelling of eddy current problems were the integral expressions for the change of the impedance of a coil placed above a conducting half-space presented by Dodd and Deeds [33–34].”
to the following version:
“A milestone in the analytical modelling of eddy current problems was the formulation of integral expressions for the change in the impedance of a coil placed above a conducting half-space, as presented by Dodd and Deeds [33–34].”

2. Lines 101 and 105: The variables r and z should be written in italics, in accordance with standard scientific notation.

3. Equations (1)–(3): The authors are kindly asked to briefly explain the meaning and physical relevance of the parameters A, P₁, and P₂, as these are not defined in the current version.

4. Lines 202–203: Please remove the excessive blank spaces that disrupt the visual flow of the paragraph.

Once these final adjustments are made, I believe the manuscript will be suitable for publication.
